# Prediction models for prostate cancer to be used in the primary care setting: a systematic review

Mohammad Aladwani,[1] Artitaya Lophatananon ![ORCID],[1] William Ollier,[1,2] Kenneth Muir ![ORCID] [1]

[1]Division of Population Health, Health Services Research and Primary Care School of Health Sciences Faculty of Biology, Medicine and Health, The University of Manchester, Manchester, UK
[2]School of Healthcare Science, Manchester Metropolitan University Faculty of Science and Engineering, Manchester, UK

**Correspondence to**
Dr Kenneth Muir;
kenneth.muir@manchester.ac.uk

## ABSTRACT

**Objective** To identify risk prediction models for prostate cancer (PCa) that can be used in the primary care and community health settings.

**Design** Systematic review.

**Data sources** MEDLINE and Embase databases combined from inception and up to the end of January 2019.

**Eligibility** Studies were included based on satisfying all the following criteria: (i) presenting an evaluation of PCa risk at initial biopsy in patients with no history of PCa, (ii) studies not incorporating an invasive clinical assessment or expensive biomarker/genetic tests, (iii) inclusion of at least two variables with prostate-specific antigen (PSA) being one of them, and (iv) studies reporting a measure of predictive performance. The quality of the studies and risk of bias was assessed by using the Prediction model Risk Of Bias ASsessment Tool (PROBAST).

**Data extraction and synthesis** Relevant information extracted for each model included: the year of publication, source of data, type of model, number of patients, country, age, PSA range, mean/median PSA, other variables included in the model, number of biopsy cores to assess outcomes, study endpoint(s), cancer detection, model validation and model performance.

**Results** An initial search yielded 109 potential studies, of which five met the set criteria. Four studies were cohort-based and one was a case-control study. PCa detection rate was between 20.6% and 55.8%. Area under the curve (AUC) was reported in four studies and ranged from 0.65 to 0.75. All models showed significant improvement in predicting PCa compared with being based on PSA alone. The difference in AUC between extended models and PSA alone was between 0.06 and 0.21.

**Conclusion** Only a few PCa risk prediction models have the potential to be readily used in the primary healthcare or community health setting. Further studies are needed to investigate other potential variables that could be integrated into models to improve their clinical utility for PCa testing in a community setting.

## INTRODUCTION

Prostate cancer (PCa) is the second most common cancer and the fifth leading cause of cancer-attributed death in men worldwide with an estimated incidence of 1 276 106 and 358 989 deaths in 2018.[1] In the UK, around 47 200 new cases of PCa were reported in

### Strengths and limitations of this study

► The review focussed on risk prediction models for PCa for use in primary care.
► The Preferred Reporting Items for Systematic Reviews and Meta-Analyses (PRISMA) approach was followed in identifying relevant articles and reporting this study.
► We used the Prediction model Risk Of Bias ASsessment Tool (PROBAST) to assess the quality and risk of bias in the included models.
► The search strategy was restricted to two databases and a manual search, to retrieve original studies.

2015, accounting for 26% of all new cancer cases in males. PCa deaths in the UK of were around 11 600 in 2016.[2] The global projections of PCa incidence and mortality for 2030 are 1.7 and 0.5 million, respectively.[3] The highest incidence of PCa is seen in western societies.[4] The significant increase of PCa incidence and diagnosis over the last three decades can be attributed mainly to the widespread implementation of the prostate-specific antigen (PSA serum test after it had been introduced in the late 1980s.[5 6]

The strong association of PSA with PCa,[7 8] along with it being a relatively inexpensive test,[9] has made PSA a key biomarker in the diagnostic process of PCa and for the recommendation of a confirmatory prostate biopsy.[7 9] PSA is, however, not a cancer-specific marker.[5 10] Conditions such as benign prostate hypertrophy (BPH), prostatitis and other non-malignant prostatic conditions can also elevate PSA level, thus introducing uncertainty to the application of the test.[11–14] This highlights limitations of the PSA test regarding its specificity and sensitivity, and it being largely dependent on setting a 'diagnostic' cut-off point, which often leads to an unacceptable number of false-positive and false-negative results.[5 10 15 16] Such issues are likely to be the part of the explanation for

the significant number of unnecessary biopsies currently being performed each year. Such procedures are associated with adverse side effects for patients and also increases healthcare costs.[17 18]

To address such PSA test limitations, researchers have incorporated other measurable factors into approaches for the early detection of PCa; these 'risk assessment tools' are based on statistical models designed to improve the accuracy and performance of the PSA test.[19–22] Logistic regression and artificial neural network (ANN) models are now considered to be the most common and effective statistical techniques in aiding the development of new models to enhance early PCa diagnosis.[23] These PCa risk prediction models can be used to aid the testing of men for further investigations.

Currently in the UK, there is no population-based screening programme for PCa. The ultimate goal of PCa screening is to find intermediate and high risk of PCa rather than low-risk PCa that would not require treatment but will give emotional burden to the patient once detected and unnecessary treatment in some patients. An important potential advantage of the extended risk models is their ability to provide a more accurate estimation of PCa risk. This may ultimately lead to their use in patient counselling and decision-making.[24–27] Such models have already achieved better results in predicting probabilities of outcome compared with clinical judgement.[28 29] Furthermore, it has been reported that using such predictive models may minimise the rate of unnecessary biopsies.[30]

Recently, there has been a substantial increase in the development of predictive models to help clinicians assess risk and decide which man to send to clinical setting to further investigate for a possible diagnosis of PCa.[22 26 30–35] The majority of these models are designed for use in clinical settings, where costs are less of an issue and most include the need for a clinical examination such as digital rectal examination (DRE) or trans-rectal ultrasonography (TRUS). One of the main limitations of DRE is its poor performance, especially at low PSA levels, and it is highly subjective to inter-observer variability.[36–38] A meta-analysis study revealed that DRE has positive predictive value of only 18%.[39] Similarly, TRUS has been reported for having poor accuracy at low PSA levels[40 41] and small PCa might not be palpable on DRE or visualisation on TRUS.[40] Furthermore, less than 40% of PCa detected by DRE are potentially curable, making it less beneficial for early diagnosis.[42] Several studies showed that there is fear, anxiety and embarrassment among some men, in particular Black men, regarding the DRE test.[43–46] Another disadvantage of the DRE is the fact that it is a potentially uncomfortable test.[47–52] This may explain why the DRE is a barrier for some men to participate in PCa screening if it includes DRE test. Lee *et al* reported that 74% to 84% of Black men may not maintain annual DRE screening,[53] while another study found that it may prevent 22% of men from participating.[54] Since TRUS needs to be performed by a skilled urologist, this means men have to make an appointment with a clinic in a different location, which makes the screening less convenient. As a result, men may feel reluctant to have such tests performed.

This systematic review of the literature was undertaken to identify risk prediction models that do not incorporate invasive or more costly clinical procedures or extensive biomarkers but have potential application for use in primary care and community settings. As low cost is a primary concern for community use, for this review, we set an indicative threshold of approximately three to five times the cost of a PSA test for inclusion. This excluded a number of models that contain new and emerging biomarker or single nucleotide polymorphism panels. As a number of persons are referred to the clinical setting, costs are less of an issue. The performance of the models reviewed for detecting PCa at initial biopsy have been compared using 'reported area under the curve' (AUC) and/or sensitivity-specificity testing.

## METHODS
The approach used to identify and select relevant articles was based on the application of the 'Preferred Reporting Items for Systematic Reviews and Meta-Analysis' (PRISMA).[55]

### Data sources and search strategy
A literature search was performed using MEDLINE (via Ovid) and Embase databases. The 'medical subject heading' (MeSH) terms, combined with Boolean logic operators 'AND' and 'OR', were applied to retrieve relevant articles. The terms used for the search were 'Prostatic Neoplasms' AND ('Initial biopsy' OR 'first biopsy' OR 'early detection of cancer') AND ('nomograms' OR 'artificial neural networks' OR 'risk assessment' OR 'statistical model'). The full search strategy is provided in a online supplementary file 1. All articles defined (published since the inception of the databases and up to the end of January 2019) were subsequently further filtered as being those only published in English language and with an abstract. Further to using the above search databases, the research articles were selected manually from the reference lists of any relevant review articles. Google Scholar and MEDLINE searches were also carried out to identify independent study for external validation for each model included in this review. The results are presented in online supplementary file 2.

### Eligibility criteria
As this review focusses on PCa risk prediction based in community healthcare settings, all studies were selected on the following inclusion criteria: (i) evaluating the risk for PCa at initial biopsy in patients who had no prior history of PCa, (ii) studies that reported 'low cost' risk assessment tools (ie, those not including more expensive genetic or biomarker test) or 'invasive' clinical tests/examinations (such as DRE or TRUS), (iii) studies that included a minimum of two variables of which PSA had

to be one of them (on the basis that an elevated PSA test in UK primary care is usually the first sign and rationale for suggesting a need for further investigation of PCa within NICE guidelines), and (iv) studies that reported AUC and/or sensitivity and specificity of the diagnostic/predictive tool. The exclusion criteria used were: (i) articles with models that were built and based on repeat or mixed biopsies, (ii) studies that only validate an existing model, and (iii) articles that were not published in English. There were no time boundaries regarding the publication year.

Screening of the titles, abstracts and full-texts was carried out by two reviewers (MA, AL). Any concerns about the eligibility of a study were resolved by discussion with a third reviewer (KM).

### Data extraction
A data extraction form was developed to collect all relevant information. For each study used in this review, the items extracted included: year of publication, source of data, type of model, number of patients, country where it was performed, age, PSA range, mean/median PSA, number of biopsy cores, variables included in the model, study endpoint(s), cancer detection, model performance and model validation.

### Evaluating the performance of the risk models
Prediction models can be evaluated against various criteria. The most critical measurements of model performance are discrimination and calibration.[27] Discrimination refers to how well the prediction model can differentiate patients in different outcome classes according to their predicted risks. It is often assessed by measuring the area under the receiver operating characteristic curve.[56] It also requires setting a series of thresholds to separate low and high ranges of predicted outcomes. A value of 0.5 indicates no discrimination, while a value of 1 indicates perfect discrimination. However, even with perfect discrimination, observed risk can differ from expected risk. Therefore, calibration has an important role in model evaluation.[57] Calibration represents the agreement between expected and observed outcomes.[58] A well-calibrated model is achieved when the calibration slope is close to 1. When the calibration slope is less than 1, it indicates that the model underestimates low risks and overestimates high risks.[59]

Due to the heterogeneity of the studies included, conducting a meta-analysis was not applicable.

### Study quality assessment
The quality of the studies included in this review was assessed using the Prediction model Risk Of Bias ASsessment Tool (PROBAST).[60] This tool has been developed specifically to assess the risk of bias and applicability for prediction model studies. The tool consists of four domains and has 20 signalling questions that facilitate reaching overall judgement of risk of bias, as well as issues relating to applicability.

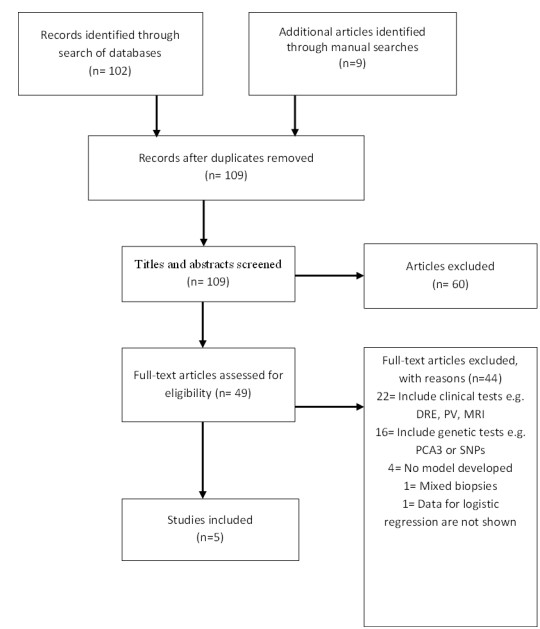

**Figure 1** Flow diagram of studies included using the Preferred Reporting Items for Systematic Reviews and Meta-Analyses method. DRE, digital rectal examination; PCA3, prostate cancer antigen 3; PV, prostate volume; SNP, single nucleotide polymorphism.

### Patient and public involvement
No patients were involved in setting the research question or the outcome measures, nor were they involved in the design and implementation of the study. There are no plans to involve patients in dissemination.

### RESULTS
A total of 102 publications were identified using the search strategy as shown in figure 1. An additional nine articles were identified through manual searches from a bibliography of reviewed articles. At the first filter step, a total of 109 titles and abstracts were screened for eligibility after removing two duplicates. In the second filter step, 60 papers failed to meet the inclusion criteria and were excluded, resulting in 49 articles. The final step of filtering yielded only five studies that were considered to be eligible (ie, passed all set criteria) and were thus included in this systematic review. There was no independent study identified for external validation for included models.

### Study characteristics
Four of the five studies included were based on cohort studies and one was a case-control study. The characteristics of each of these studies and populations are summarised in table 1. Details of PSA assays used in the models are presented in online supplementary file 3.

Patients used to build the risk models varied across these studies. Of the five studies, three studies included men from referral populations[61–63] and two studies from screening programmes.[64 65] The sample sizes ranged from 151 to 3773, with three studies derived from US

**Table 1** Characteristics of the included studies

| Author and year | Type of model | Type of study | Sample no. | Location | Population type | Age | Median age | PSA range | Median PSA | No. of biopsy cores | Cancer detection |
|---|---|---|---|---|---|---|---|---|---|---|---|
| Carlson et al, 1998[61] | Logistic regression | Cohort | Model dev=3773 Validation=525 | Baltimore, USA | Referral | ≥45 | — | 4 to 20ng/mL | — | Sextant biopsy | 32% |
| Babaian et al, 2000[64] | Neural network=3 ANNs | Cohort | 151 | Texas, USA | Screening programme | 40 to 75 | 62 | 2.5 to 4ng/mL | — | 11 cores | 24.50% |
| Jansen et al, 2010[65] | Logistic regression | Cohort | Site 1=405 | Site 1 from the Rotterdam arm of the European Study of screening for Prostate cancer | Screening programme | ≥50 | Site 1 (66) | 2 to 10ng/mL | ~4.4 | ≥6 cores | Site 1=55.8% |
|  |  |  | Site 2=351 | Site 2 Innsbruck, Austria |  |  | Site 2 (60) |  |  |  | Site 2=49.6% |
| Hill et al, 2013[62] | Logistic regression | Case-control | 1378 | Florida, USA | Hospital referral | 40 to 90 | — | ≥4ng/mL | — | N/A | 20.60% |
| Lazzeri et al, 2013[63] | Logistic regression | Cohort | 646 | European multicentre; Italy, Germany, France, Spain, and the UK | Referral | >45 | — | 2 to 10ng/mL | ~5.8 | ≥12 cores | 40.10% |

ANN, artificial neural network ; PSA, prostate-specific antigen.

**Table 2** Variables used in the prostate cancer risk prediction models

| Author and year | Variables used in the model | | | | |
|---|---|---|---|---|---|
| | Total PSA | Free PSA | Per cent free PSA | Age | Other variables |
| Carlson et al, 1998[61] | ✓ | | ✓ | ✓ | |
| Babaian et al, 2000[64] | ✓ | ✓ | | ✓ | Creatine kinase, prostatic acid phosphatase |
| Jansen et al, 2010 (Site 1)[65] | ✓ | ✓ | | | p2PSA |
| Jansen et al, 2010 (Site 2)[65] | ✓ | ✓ | | | p2PSA |
| Hill et al, 2013 (Method 1)[62] | ✓ | | | ✓ | HGB, RBC, haematuria, creatinine, MCV and ethnicity 'Black' |
| Hill et al, 2013 (Method 2)[62] | ✓ | | | ✓ | HGB, RBC, creatinine and MCV |
| Lazzeri et al, 2013 (Model 1)[63] | ✓ | ✓ | ✓ | | |
| Lazzeri et al, 2013 (Model 2)[66] | ✓ | ✓ | ✓ | | p2PSA |
| Lazzeri et al, 2013 (Model 3)[63] | ✓ | ✓ | ✓ | | %p2PSA |
| Lazzeri et al, 2013 (Model 4)[63] | ✓ | ✓ | ✓ | | PHI |

HGB, haemoglobin; MCV, mean corpuscular volume; PHI, prostate health index; p2PSA, precursor of PSA; %p2PSA, percentage of p2PSA to free PSA ; PSA, prostate-specific antigen; RBC, red blood cells.

cohorts[61 62 64] and two from Europe.[63 65] Four studies used logistic regression methodology to build their model, whereas one study used an ANN-based approach.[64] The minimum age of participants was 40 years[62 64] and the minimum PSA level was 2 ng/mL.[63 65]

### Variables in the model

Table 2 presents details of the variables used in each model. PSA level was used in all models, followed by free PSA (fPSA), age and free-to-total PSA ratio (%fPSA). Other variables also reported in the models included: precursor of PSA (p2PSA), percentage of p2PSA to fPSA (%p2PSA), prostate health index (PHI), levels of haemoglobin (HGB), albumin, creatinine and red blood cell count (RBC), haematuria, mean corpuscular volume (MCV) and prostatic acid phosphatase.

### Outcome

The study endpoint also varied among the studies selected. Two studies evaluated the accuracy of detecting any PCa[61 64] and three studies examined the pathologic Gleason score.[62 63 65] Although Jansen et al, did not build a model to predict the aggressiveness of PCa, they assessed the relationship of each variable individually with a Gleason score ≥7. PCa was determined by taking a needle biopsy. All patients in the five studies underwent prostate biopsy. The least number of biopsy cores used were six[61 65] and the highest were ≥12.[63] One study did not report the number of biopsy cores taken.[62] PCa rates ranged from 20.6% to 55.8%.

### Evaluating the performance of the risk models

For predicting any PCa, the Jansen et al, model used data from the Rotterdam arm of the European Study of Screening for PCa (ESPRC). Their model achieved the highest discrimination value when compared with PSA

alone (AUC of 0.755 vs 0.585, respectively).[65] The AUC values in other studies ranged from 0.648 to 0.74.

One study did not provide the AUC but instead reported an increase of 11% in specificity over per cent fPSA alone with 95% sensitivity.[61] Lazzeri et al,[63] presented results from four separated models discriminating PCa with a Gleason score of ≥7. Lazzeri's model 2 (which includes the base model total PSA, fPSA and %fPSA in addition to p2PSA) and model 3 (which includes base model plus PHI) showed the highest levels of discrimination out of the four models with an AUC of 0.67. In the study of Hill,[62] the authors classified PCa stages differently and built their two models accordingly. In Hill's first model, the difference in the discrimination was analysed and based on all PCa versus non-cancerous prostate conditions where the AUC for this model was 0.68 compared with 0.59 for PSA alone. In Hill's second model, the discrimination analysis was based on PCa stages II, III, IV versus PCa stage I, prostatic interstitial neoplasm, BPH and prostatitis where stages I, II, III and IV are parallel to T1, T2, T3/T4 and metastatic PCa, respectively. The AUC for the second model was 0.72 compared with 0.63 for PSA alone. In general, four studies examined the AUC with PSA alone and all reported a benefit from the use of logistic regression or the trained ANN. Model performance and the differences between the AUC's for PSA alone and for the extended models are presented in table 3.

Sensitivity and specificity data are presented in table 4. At 95% sensitivity, the Babaian et al model shows the highest specificity (51%) whereas the Jansen model for both sites had the lowest specificity (~23.5%). In the Hill study, with a sensitivity of ~90%, the specificity was lower than in other studies (~18% and 28%) for method 1 and 2, respectively. In the study reported by Lazzeri, the sensitivity and specificity were not reported for the overall

**Table 3** The difference of AUC for PSA alone and extended model

| Study | AUC for PSA | AUC for model | ΔAUC (Model – PSA) |
|---|---|---|---|
| Carlson et al[61] | NA | NA | NA |
| Babaian et al[64] | 0.64 %fPSA | 0.74 | 0.1 |
| Jansen et al (Site 1)[65] | 0.58 | 0.75 | 0.17 |
| Jansen et al (Site 2)[65] | 0.53 | 0.7 | 0.16 |
| Hill et al (Method 1)[62] | 0.59 | 0.68 | 0.09 |
| Hill et al (Method 2)[62] | 0.63 | 0.72 | 0.09 |
| Lazzeri et al[63] | 0.50 for any PC 0.54 for Gleason score ≥7 | Model 1=0.65 Model 1 (Gleason score ≥7)=0.60 | 0.15 0.06 |
| | | Model 2=0.71 Model 2 (Gleason score ≥7)=0.67 | 0.21 0.13 |
| | | Model 3=0.704 Model 3 (Gleason score ≥7)=0.67 | 0.2 0.13 |
| | | Model 4=0.71 Model 4 (Gleason score ≥7)=0.672 | 0.21 0.13 |

AUC, area under the curve; %fPSA, free-to-total PSA ratio ; PC, prostate cancer; PSA, prostate-specific antigen.

model; instead their study reports sensitivity and specificity for predictive variables individually. The highest sensitivity (90.5%) of %p2PSA and %fPSA achieved the

highest specificity in predicting PCa at 21.5% and 22.8%, respectively. Percentage p2PSA and PHI were more associated with Gleason scores.

Table 5 summarises the validation and calibration results for the studies included. Model calibration was reported in two studies.[61 63] Carlson plotted the observed and expected risks using calibration plots, whereas Lazzeri used the Hosmer-Lemeshow goodness-of-fit test. In terms of validation, two studies did not report model validation.[62 65] Only one study reported an external validation using an additional data set consisting of 525 patients.[61] Cross-validation using multiple re-sampling schemes was used in the Babaian study; however, they did not report the number of time this was performed.[64] Lazzeri used 200 bootstrap re-samples to minimise overfitting bias.[63]

**Study quality assessment**

Quality assessment was carried out by two reviewers (MA and AL) with any discordance resolved by a third reviewer (KM). The assessment of results suggested some issue of study quality according to the criteria as set in the PROBAST, particularly in the analysis domain. For instance, one study applied univariable analysis to select predictors.[61] Three studies did not measure calibration.[62 64 65] Furthermore, two studies did not account for optimism and overfitting by using internal validation methods.[62 65] Whereas one study did not use appropriate measures for model performance that is, AUC, this study reported the calibration.[61]

The event per variable was lower than recommended (<10)[59 66] in the Babaian[64] study, indicating inadequate

**Table 4** Sensitivity and specificity profile at different levels for each model*

| Study | Sensitivity | Specificity | Probability cut-off | Positive predictive value | Negative predictive value |
|---|---|---|---|---|---|
| Carlson et al[61] | 99 | 18 | >15 | ≤47 | NA |
| | 95 | 34 | 18 | 51 | NA |
| | 89 | 43 | 20 | 42 | NA |
| Babaian et al[64] | 95 | 51 | NA | 39 | 97 |
| | 92 | 62 | NA | 44 | 96 |
| | 89 | 62 | NA | 43 | 95 |
| Jansen et al (Site 1)[65] | 95 | 23.9 | NA | NA | NA |
| Jansen et al (Site 1)[65] | 90 | 30.1 | NA | NA | NA |
| Jansen et al (Site 2)[65] | 95 | 23.2 | NA | NA | NA |
| Jansen et al (Site 2)[65] | 90 | 36.2 | NA | NA | NA |
| Hill et al (Method 1)[62] | 90.9 | 17.6 | 33 | 47.1 | 70.5 |
| Hill et al (Method 2)[62] | 89.8 | 28 | 13 | 20.6 | 91.3 |
| Hil et al (Method 1)[62] | 80.5 | 37.1 | 37 | 50.9 | 70.2 |
| Hill et al (Method 2)[62] | 78.2 | 45 | 15 | 28.7 | 88.8 |
| Hill et al (Method 1)[62] | 39.9 | 81.4 | 48 | 63.4 | 62.6 |
| Hill et al (Method 2)[62] | 45.8 | 79.5 | 23 | 36.7 | 85 |

*Lazzeri[63] model reported only sensitivity and specificity for predictive variables individually and at sensitivity of 90, %p2PSA and %fPSA achieved the highest specificity
%fPSA, free-to-total PSA ratio ; NA, not applicable; %p2PSA, percentage of p2PSA to free PSA .

**Table 5** Validation and calibration for included models

| Author and year | Validation | Calibration |
|---|---|---|
| Carlson et al,1998[61] | External validation on additional data set consisting of 525 patients | Calibration plot |
| Babaian et al, 2000[64] | Cross-validation and separate data set of 151 | NA |
| Jansen et al, 2010[65] | NA | NA |
| Hill et al, 2013[62] | NA | NA |
| Lazzeri et al, 2013[63] | Internal validation using 200 bootstrap resamples | Internal calibration using the Hosmer-Lemeshow goodness-of-fit test |

power. Four studies did not report missing data or how they handled it.[61 63–65] The remaining study used complete case analysis and excluded patients with missing data on laboratory biomarkers (n=75).[62] The PROBAST guidelines state that in a prediction model study where any risk of bias and applicability is low in all four domains, a regrading to high risk of bias should be considered when the study did not validate the model externally.[60] Thus, although the quality assessment for the Lazzeri study[63] was graded low risk in all the four domains, since the study did not report any external validation of the model, the assessment of the study has been regraded to high risk of bias according to the PROBAST criteria. A full quality assessment for all studies is presented in table 6.

## DISCUSSION

Despite the large number of PCa risk prediction models, the majority still include clinical inputs and/or more costly biomarker or genetic panels; few low cost models exist that do not include specialist clinical input or more expensive further testing that limits there use for population wide assessments. To our knowledge, this is the first study to examine risk prediction models for PCa that are low cost and do not include clinical and genetic variables, and are based on single time-point assessment.

Our study identified five unique models that met the set criteria. The Carlson model[61] has the largest population (3773 patients) when compared with the other four studies. Although they reported an 11% increase in specificity, they did not report AUC predictive estimates. It has been acknowledged that sensitivity and specificity results

are dependent on the prevalence of the disease. Hence, the comparison between populations where the PCa prevalence may vary (especially in early detection) will be difficult.[33] More importantly, by not reporting the AUC estimate, the model raises some doubts regarding the reliability of the model and its implementation.[33] It will also make comparison to other models not applicable.[67]

Babaian[64] developed an algorithm and compared the performance of the ANN to PSA density (total PSA divided by prostate volume) (PSAD), %fPSA and transition zone density (PSAD-TZ). Their ANN demonstrated a significant increase of model specificity that reached 51% when sensitivity was held at 95%. This was better than the specificity value of each individual variable such as %fPSA (10%), PSAD (39%), and PSAD-TZ (22%). In terms of AUC, the ANN achieved a moderate accuracy (0.74), being the second highest among all studies included. However, the ANN model did not show significant improvement when compared with a model fitted with only individual variables (AUC for %fPSA=0.64, PSAD=0.74 and PSAD-TZ=0.75). They included a number of uncommon pre-biopsy inputs into their algorithm such as prostatic acid phosphatase and creatine kinase.[68] Furthermore, they used a tight PSA range (2.5 to 4.0 ng/mL) which meant that their model may be less suitable for patients with PSA level below or above that range, thus limiting its generalisability.

The study by Jansen and colleagues[65] demonstrated that adding p2PSA to the base model of PSA and fPSA significantly enhanced the PCa predictive value and specificity. The association and added value of p2PSA in the

**Table 6** Quality assessment for ROB and applicability concern for included studies

| Study | ROB* | | | | Applicability | | | Overall | |
|---|---|---|---|---|---|---|---|---|---|
| | Participants | Predictors | Outcome | Analysis | Participants | Predictors | Outcome | ROB | Applicability |
| Carlson et al[61] | + | + | – | – | + | + | – | – | – |
| Babaian et al[64] | + | + | + | – | – | + | + | – | – |
| Jansen et al[65] | – | – | – | – | + | – | – | – | – |
| Hill et al[62] | – | + | + | – | – | + | + | – | – |
| Lazzeri et al[63] | + | + | + | + | + | + | + | – | + |

+ indicates a ROB or applicability; – indicates a high ROB or applicability.
*ROB, risk of bias.

prediction and detection of PCa have been reported by several other studies.[16 69–72] Jansen[65] showed that p2PSA has no clear association with aggressive PCa. However, the base model that includes p2PSA had the highest clinical significance in correlation to pathologic Gleason score with a p value of 0.008 compared with %fPSA and PHI (p value 0.01 and 0.02, respectively). Although they used archived blood samples and retrospective analysis, the results were similar to a prospective study of 268 patients.[16]

Hill[62] used a case-control study to evaluate several laboratory biomarkers. They found HGB, RBC, haematuria, creatinine, PSA, age, MCV and ethnicity ('being Black') were statistically significantly associated in the first method (p<0.05). In the second method, HGB, RBC, creatinine, PSA, age and MCV were found to be statistically significantly correlated (p<0.001) with PCa. However, since this study was designed as a case-control study, it would have been more prone to uncontrolled confounding and selection bias. Moreover, the type of screening protocols used in Veterans' Administrations may vary to those conducted in other healthcare systems; therefore, the results may not be applicable to other populations. Furthermore, patients with a PSA level <4.0 ng/mL have not been investigated, and thus, the performance of the models are unknown for individuals in this group.

Lazzeri *et al*,[63] in a European multicentre study have evaluated similar biomarkers as in Jansen study with the same PSA range 2 to 10 ng/mL prospectively. They found no difference in both %p2PSA and PHI as individual PCa predictors with AUC of 0.67 (95% CI 0.64 to 0.71). However, the base model (consisting of PSA, fPSA and %fPSA) that also included either p2PSA or PHI outperformed the base model alone and the base model including %p2PSA. In the analysis, the additive value of both p2PSA and PHI is 0.064 and 0.056 for %p2PSA for predicting the risk of PCa. These additive values increased to 0.076 for both p2PSA and PHI, and 0.073 for %p2PSA in predicting Gleason scores ≥7 for the disease. The usefulness of PHI in improving the predictive accuracy of PCa over total and free PSA has been confirmed and reported by several studies.[16 72–75]

In general, only one study has validated their model externally,[61] whereas the remaining studies were either validated internally[63 64] or did not report any validation methods.[62 65] Prediction models may not be equally applicable to all data sets as patients' characteristics may vary.[20 76] As a result, the generalisability of a model might be poor when it used in populations other than that used in building the model. Therefore, external validation should be conducted before applying any new model into general practice.[77 78]

Another key performance measure of any model that needs careful evaluation is calibration. A calibration plot with a calibration slope is more preferable than the Hosmer-Lemeshow test; it has been acknowledged that evaluating a good and well calibrated model based on a large data set can still fail the Hosmer-Lemeshow test. In contrast, when evaluating a poorly calibrated model with a small data set it can still pass the Hosmer-Lemeshow test.[79] In our analysis, three studies fail to report the calibration of the model[62 64 65] while the Carlson study[61] used a calibration plot and Lazzeri[63] used the Hosmer-Lemeshow test. Excluding calibration from the majority of models may explain why some models are not currently used in practice.[79]

With regard to biopsy cores, only two studies used extended biopsy cores. Babaian[64] used an 11-core multisite biopsy, whereas Lazzeri[63] used at least 12 biopsy cores. Moreover, two studies used six cores biopsy in their model.[61 65] The use of six-core biopsy has been criticised as not being adequate in detecting PCa[80] and that models developed using sextant biopsy are less accurate than when a 10-core biopsy is used.[76] As a result, the European guideline for clinical PCa recommended an extended biopsy as standard practice for PCa detection.[81]

It is worth noting that all five reviewed models performed better than just PSA alone. However, none of them has both high specificity and sensitivity. The level of sensitivity has been increased, and despite enhancement in the specificity, it is still considered low. Specificity is crucial when it comes to being used in a population setting as men without PCa should be ruled out as much as possible from further invasive engagement with the health system.

Our review therefore suggests that none of the reviewed models provide an ideal performance in predicting PCa with high sensitivity and high specificity. It is particularly important when considering the application of PCa risk prediction at the population level that the tool used should be able to both detect the outcome and filter out people with no disease. As there is robust evidence suggesting the clinical relevance of PSA range to the detection of PCa differs across age groups,[82–84] any future model should consider PSA threshold in relation to a specific age range. Risk prediction models for PCa should therefore take account of age.

Out of the five reviewed models, the Lazzeri model 2, has the greatest potential to be implemented in primary care. It achieved the least risk of bias and had fair discrimination for both any and aggressive PCa. It also had the largest improvement in discrimination performance compared with PSA alone. Moreover, except for the p2PSA that requires a specific assay, the included variables are common and easy to measure. However, before it could be used, the model requires to be validated externally.

## Comparison with other studies
To our knowledge, three systematic reviews of PCa prediction tools have been published.[20 26 27] In the Louie *et al* review, risk models were included that were externally validated in at least five study populations for the purpose of meta-analysis and only six studies were included in their analysis. Furthermore, all the studies included incorporated clinical tests such as DRE and/or TRUS-PV.[26] Schroder and Kattan[20] reviewed models that were

built to predict the likelihood of having a positive prostate biopsy for cancer. However, it appears that they also included models where patients had a previous negative biopsy. As such, some of the models included variables related to biopsy results and cores. The review by Shariat and colleagues examined different types of predictive tools.[27] They explored tools that predict PCa on initial and repeat biopsy, pathologic stages, biochemical recurrence after radical proctectomy, metastasis, survival and life expectancy. Similarly, virtually all the prediction tools that were based on initial biopsy included variables based on invasive procedures.

### Strengths and limitations of this study

This report is the first to review risk prediction tools for PCa that can be used in primary care and community settings. Any prediction model should therefore be simple to use, based on non-invasive tests, be feasible at a population level and at low cost. We carried out an extensive data extraction relating to important features and characteristics for each study included, such as modelling method, source of data, sample size, variables, discrimination, validation and cancer detection rate. We have also followed PRISMA guidelines for identifying eligible articles as well as for reporting this study. In addition, the PROBAST was adopted to assess the quality and risk of bias for each prediction model.

Our study has some limitations. Our aim was to identify prediction models that have the potential to be implemented in a primary care or community setting, and consequently our search strategy was to retrieve relevant studies for this specific purpose. Furthermore, we excluded articles that were not published in English or did not have an abstract. Moreover, only two databases were searched, besides manual search, to retrieve original studies.

A previous systematic review suggested that the majority of relevant studies could be identified through a manual search of articles reference lists instead of a database alone.[20] We identified four eligible studies using this approach. Given the small number of models identified by the approach we followed, that can be applied in primary care settings compared with the large number relating to wider existing models, it is unlikely that we have not included any study that would affect the results of our review.

### Implications and future research

It is now accepted that the PSA test and its derivatives have some limitations for detecting PCa as defined by subsequent biopsy.[85] As a consequence, a considerable number of PCa prediction models have been built to improve prediction accuracy. This has resulted in a plethora of PCa risk prediction tools, with to date more than 100 models described.[86 87] There is evidence that some of these models show benefit and have better performance over just PSA measurement alone.[20] It also has been demonstrated that some of these models out-performed clinical

experts in predicting PCa.[28 29] Although such models are not designed to replace specialist clinical judgement or patient preferences,[76 85 88] they can help in patient counselling and aid clinicians to decide whether a prostate biopsy should be taken or not.[77 88 89]

Given the small number of risk prediction models for PCa that do not incorporate clinical or genetic tests, the discrimination of these reviewed models ranged between poor to moderate (AUC range ~0.65 to~0.75); in addition there were some issues relating to their study design and analysis raises the risk of bias. Consequently, none of these models could be currently recommended for use in a primary care and community healthcare setting. Several guidelines are against using PSA test based screening for PCa; the US Preventive Services task force, the Canadian task force on preventive health and the American College of Preventive Medicine do not currently recommend PSA-based testing due to insufficient evidence.[90–92] This has made it difficult, so far, to convince policymakers to adopt PCa screening programme.

The first guideline of PROSTATE CANCER UK states, "In the future, health professionals should look at a man's PSA level alongside other known risk factors as part of a risk assessment tool, when one becomes available."[93] However, the vast majority of the current PCa risk prediction models are not suitable for routine use as they include clinical and genetic tests and are not validated externally in other cohorts. Therefore, the main challenge in the UK, remains to develop a risk prediction tool that is reliable, cheap, is applicable for as wide an ethnicity as possible, and, most importantly, is easy to use and can be implemented at a primary care level.[94]

The value of such risk tools is that they will help to stratify men at high risk of developing PCa earlier so that they have appropriate management and/or surveillance programme as early as possible and, therefore, may fit into the clinical pathway. Such tools should help physicians have a better understanding of the risk for this disease and simplify the procedures and discussions with patients when recommending further specialist-led investigations such as DRE and/or MRI where a decision on whether a biopsy should or not perform is concluded. Furthermore, using the appropriate risk prediction tool will avoid men from undergoing inappropriate further and frequent testing.[94] This will reduce any associated costs of inappropriate tests and decrease the burden on healthcare delivery systems.

It is crucial to address these issues by identifying all possible risk factors for PCa that are non-clinical, non-genetic, and easy to use and interpret. There remains a pressing need to develop a risk prediction tool in the future using all appropriate factors (potentially also including genetics once there is infrastructure in place for genetic testing in the primary care and the cost comes down) into a robust multivariable analysis and validate the model externally to eliminate applicability and generalisability concerns. Only when this is achieved will it be

possible to introduce a PCa screening programme fit for purpose.

## CONCLUSION

There is a paucity of suitable low-cost risk models that incorporate non-clinical, non-genetic inputs and which can be used at a primary care level and in other community health services. Existing models have limitations reflecting both study design and reporting performance measures. Future research should take into account these key issues and explore other risk factors for incorporation into further models.

**Acknowledgements**  MA is supported by Kuwait's Ministry of Health.

**Contributors**  MA, AL and KM were involved in study conception, idea and design. Data acquisition and extraction was obtained by MA. Data synthesis and interpretation was carried out by MA, AL and WO. MA and AL drafted the manuscript. All authors approved the final version of manuscript. KM is the study guarantor.

**Funding**  KM and AL are supported by the NIHR Manchester Biomedical Research Centre and by the ICEP (This work was also supported by CRUK (grant number C18281/A19169)). KM is also supported by the Allan Turing Institute.

**Competing interests**  All authors have completed the ICMJE uniform disclosure form at http://www.icmje.org/coi_disclosure.pdf and declare: no support from any organisation for the submitted work; no financial relationships with any organisations that might have an interest in the submitted work in the previous 3 years; no other relationships or activities that could appear to have influenced the submitted work.

**Patient and public involvement**  Patients and/or the public were not involved in the design, or conduct, or reporting, or dissemination plans of this research.

**Patient consent for publication**  Not required.

**Provenance and peer review**  Not commissioned; externally peer reviewed.

**Data availability statement**  Data are available upon reasonable request. Data extraction sheet and the PROBAST assessment form for risk of bias is available upon request.

**ORCID iDs**
Artitaya Lophatananon http://orcid.org/0000-0003-0550-4657
Kenneth Muir http://orcid.org/0000-0001-6429-988X

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
