## [Reviewer comments · BMJ Open]

ARTICLE DETAILS

TITLE (PROVISIONAL)	Prediction models for prostate cancer to be used in the primary care setting: a systematic review.
AUTHORS	Aladwani, Mohammad; Lophatananon, Artitaya; Ollier, William; Muir, Kenneth

VERSION 1 – REVIEW

REVIEWER	Matthew Roberts University of Queensland Australia
REVIEW RETURNED	06-Nov-2019

GENERAL COMMENTS	This paper sought to identify and review risk prediction models that did not include invasive clinical or genetic information. The review is an interesting approach, given the patient discomfort of invasive clinical procedures- their questionable diagnostic accuracy and cause for unnecessary biopsies in non-MRI based pathways should also be noted. There are some issues that affect the foundation of the paper's aims and methods I think that require addressing, see below. Major comments: 1. With multiparametric MRI gaining such wide adoption, prior to initial biopsy predominantly based on UK studies, and I believe being implemented in many NHS Trusts, should MRI be considered in the research question as a "non-invasive" clinical test? This would obviously change the article selection criteria and search strategy, but should be considered to be most relevant for contemporary practice.2. I have significant concerns that the search strategy is too limited. For a search term of PSA-based risk stratification, to only get approx. 100 articles seems very low. I think a wider ranging search strategy is required, as that listed is not what is typically listed in a Cochrane review or other high impact review.a. For instance, the use of ("Nomograms" OR "Artificial neural networks" OR "Risk assessment" OR "Statistical model") may be particularly restricting. For example, the paper by Vickers et al PMID: 19682790 I would have thought should be eligible, but does not mention any of these terms. There may be many more.b. I would suggest the authors consult the Cochrane guidelines and/or librarian skilled in systematic literature review strategy to widen the search and increase potential article selection. While I appreciate it is time consuming to read through large numbers of abstracts, this is a necessary process to ensure important articles are not missed.
---

	3. While the authors state that the PRISMA statement was followed, there are some major omissions that should be part of a high quality systematic review. a. The date of the search and exact combination of search terms for each search engine should be specified. In the current state, I would be unable to replicate the study. b. How was Medline accessed? Through OVID, Web of Science or other? c. Why were the articles limited to English? This is becoming an increasingly unjustifiable reason for exclusion, and many non-English speaking countries provide good studies given limited resources. Expanding this is worth considering. d. Screening of articles was only performed by one reviewer – this should be performed by two reviewers, with discrepancies resolved after discussion or consultation of a third party. 4. Statistically, it is stated “Due to the heterogeneity of the studies included, conducting a meta-analysis was not applicable.” However, there appears to be a prevalence meta-analysis performed for the AUC estimates. I am not sure how valid the multiple comparisons of the same databases/study populations with different variables is. I presume from the author affiliations that they are skilled in statistical principles, however if not, then perhaps consultation with a statistician with expertise in meta-analysis is warranted. a. If prevalence or other meta-analysis was performed, what parameters and software was used? b. Further to this, potential subgroup analysis may allow better explanation of heterogeneity (cancer detection rates 20-50% is a huge variation) 5. A key evolution from the screening trials and risk stratification is clinically significant cancer detection rate, can this also be provided for each study? This really is the key in guiding management. While I appreciate that the likely target here is the wider medical community and general practitioners/primary care physicians, perhaps it is worth the authors consulting an academic urologist to address the nuances in PSA-based prostate cancer detection and management. Minor comments: 6. The introduction is appropriate in scope and length given the target general audience of BMJ Open 7. Were the PSA assays used the same? There are many different assays that can cause inter-test variability, so may be worth listing in a demographic or supplementary table 8. Table 1 – median age should also be included. 9. Page 12 Lines 31-53 is difficult to follow 10. Page 16 Lines 7-9 “The event per variable (EPV) was lower than recommended (< 10) in the Babaian 44 study indicating inadequate power.” – is this a valid measure and if so, please provide supporting literature 11. As BMJ Open is an online journal, I would suggest colour coding in addition to the +/- for Table 6
--	---

REVIEWER	Robert W Foley University Hospitals Bristol NHS Foundation Trust, United Kingdom
REVIEW RETURNED	07-Nov-2019

GENERAL COMMENTS	Overall, nice paper.
----------------------

However, I think the authors need to be very careful about the message they are sending with this paper. The authors want to look at a 'non-clinical' risk calculator that does not include 'invasive' testing, and include the digital rectal examination (DRE) in this invasive category. They seem to justify this statement in the introduction because the DRE may cause anxiety, reference number 36. This reference actually does not state that the DRE may cause anxiety, in fact it states the opposite and references an earlier study demonstrating the DRE does NOT cause additional anxiety. This sentence in the introduction should be removed.

This also raises the greater point of the use of DRE in prostate cancer risk stratification. It is essential that a DRE is performed in patients under investigation for prostate cancer, to suggest otherwise would not be good practice, which is implied by this study not including risk models using DRE. I think the only way to justify this study is to talk about the difficulty of an accurate DRE in the community setting, i.e. not from a urologist. I do not know the literature on this, but I think the authors need to think carefully about this before re-writing the introduction to justify not including DRE as a risk factor in a PCa risk model.

I feel the addition of a urologist to the paper, although not essential, would add a clinical opinion, which would be of great benefit to this work.

I feel the labelling of TRUS as invasive makes sense, unlike DRE. However, it is also important to note that TRUS is not performed as an assessment tool in clinical practice and so any model including TRUS-derived variables is quite out of date. TRUS is now only used to guide biopsy, and hence the decision for biopsy has already been taken.

Methods:

The authors do not state the statistical software and method used to create Figure 2, please add to methodology.

The authors state that external validation is very important for a risk model and include this in the paper. "External validation was only carried out in one paper". However the authors have chosen not to include external validation studies in the analysis. How do the authors reconcile this? If there are external validation studies of the models included in this paper then the external validation studies should be mentioned.

Discussion:

Very thorough discussion.

Please include in the limitations section that because you are searching for a non-clinical risk model for PCa, none of the models which include the PHI score are suitable as this would mean the addition of a biochemical test that is not routinely performed in clinical practice. This would have additional cost implications.

I do feel as if the authors do not fully understand the management pathway of a patient under investigation for PCa. The use of a

	'non-clinical' PCa risk calculator may fit into the clinical pathway, but the point it would fit in would have to be in the community to aid in the decision to refer a patient to a urologist. At which point improved risk stratification could then be undertaken following a DRE. The decision would then be made on whether to proceed to MRI, proceed directly to biopsy or to not perform biopsy. If an MRI is performed, the decision would then be made on prostate biopsy. Please include in the intro/discussion (as appropriate) where in the pathway a non-clinical model may be of use. The authors have recommended that a non-clinical risk model be made, and externally validated etc. Stating that "It is crucial to address these issues by identify all possible risk factors for PCa that are non-clinical, non-genetic, and easy to use and interpret." However, I am not sure there is a clinically unmet need for a "non-clinical" risk calculator that does not include DRE, so the authors will need to justify this with evidence from the literature (and perhaps the input of general practice and urology colleagues).
--	---

REVIEWER	L.G.W. Kerkmeijer, MD, PhD Radboud UMC Nijmegen, The Netherlands
REVIEW RETURNED	18-Nov-2019

GENERAL COMMENTS	Well written paper and analyses. Please add more clinical background to the introduction and discussion. Please describe the limitations in DRE and TRUS. Please describe the additional role of multiparametric MRI in an initial phase of prostate cancer detection (after PSA, before biopsy), as this is non-invasive and non-genetic (and falls within the description of the title). Although mpMRI has obviously no role in the primary health care setting, but has a role in the hospital setting. Please mention the ultimate goal of prostate cancer screening: to find intermediate and high risk prostate cancers (and not low risk prostate cancers that would not require treatment, but will give emotional burden to the patient once detected and generate unnecessary treatment in patients not coping with active surveillance treatment). Please adapt your title to something like: 'prediction models for prostate cancer for primary health care: a systematic review', as this may better reflect the purpose and suggested use of the outcome of the review.
---

REVIEWER	Mark Clements Karolinska Institutet Sweden I am an investigator on the STHLM3 diagnostic trial. I do not have any financial competing interests to declare.
REVIEW RETURNED	04-Feb-2020

GENERAL COMMENTS	+ This is a very nicely written article - I appreciated the opportunity to read it. The review was carefully undertaken and thoughtfully interpreted. I particularly enjoyed the Discussion and Conclusions, where the authors seem to be unconvinced by the current evidence on the available tests for community-based testing. + My main concern is whether the article adds to the literature. In particular, there is so much heterogeneity between the studies, including PSA thresholds, study cohort definitions and biopsy protocols, that comparisons between the studies becomes
---

	increasingly uninformative. I realise that the change in AUCs is trying to provide more valid internal contrasts, however study heterogeneity may also be associated with those deltas. I was less convinced about the importance of the population being biopsy-naive -- particularly when the current populations will increasingly have a large proportion of men having had a previous biopsy. Moreover, I was unclear why genetics should not be routinely included in community screening, particularly if such tests come down in cost. The authors may care to comment on these issues. + There is an interesting issue with prostate cancer testing: should it be framed in terms of community testing or in terms of a clinical diagnostic pathway? For example, community-based testing should be inexpensive, however the choice of "screening" test affects *who* will be referred to a urologist, who may undertake an MRI, and may include further clinical information (e.g. second PSA test value, DRE, or prostate volume) to decide whether to undertake a biopsy. Decisions earlier in this testing and diagnostic pathway have resource and predictive implications for later in this pathway. The authors have set themselves firmly at the start of this pathway. Do these issues affect the interpretation of their findings? + Remarkably, I have little to criticise for the existing manuscript. Some of the English language could be improved. I have the following minor points:  - I was unclear about the interpretation of the phrase "evaluating the prospective risk" - please re-phrase for clarity. - The search terms seemed somewhat specific - what was the rationale for those terms, please?
--	---

VERSION 1 – AUTHOR RESPONSE

Reviewer #1:

Comment 1:

- This paper sought to identify and review risk prediction models that did not include invasive clinical or genetic information. The review is an interesting approach, given the patient discomfort of invasive clinical procedures- their questionable diagnostic accuracy and cause for unnecessary biopsies in non-MRI based pathways should also be noted.
- There are some issues that affect the foundation of the paper's aims and methods I think that require addressing.

Response:

- We thank the reviewer for taking time to read the manuscript and finding it interesting. The reviewer comments are addressed accordingly.

Comment 2:

- With multiparametric MRI gaining such wide adoption, prior to initial biopsy predominantly based on UK studies, and I believe being implemented in many NHS Trusts, should MRI be considered in the research question as a "non-invasive" clinical test? This would obviously

change the article selection criteria and search strategy, but should be considered to be most relevant for contemporary practice.

Response:

- We agree that MRI is increasingly being used in diagnosis of PCa. However, it is still limited resource in primary care settings and this article aims to review prostate cancer risk prediction that can be used in the primary care/community setting that will aid flagging up anyone who would require further examination including MRI procedures. This will filter out men and not overload the health system for undergoing MRI.

Comment 3:

- I have significant concerns that the search strategy is too limited. For a search term of PSA-based risk stratification, to only get approx. 100 articles seems very low. I think a wider ranging search strategy is required, as that listed is not what is typically listed in a Cochrane review or other high impact review.
 - a. For instance, the use of (“Nomograms” OR “Artificial neural networks” OR “Risk assessment” OR “Statistical model”) may be particularly restricting. For example, the paper by Vickers et al PMID: 19682790 I would have thought should be eligible, but does not mention any of these terms. There may be many more.
 - b. I would suggest the authors consult the Cochrane guidelines and/or librarian skilled in systematic literature review strategy to widen the search and increase potential article selection. While I appreciate it is time consuming to read through large numbers of abstracts, this is a necessary process to ensure important articles are not missed.

Response:

- We have discussed the pros and cons of our search approach in strengths and limitations section and believe that this adequately describe the process we followed which has merit. Also, we have used search terms similar to two previous systematic reviews in PCa risk prediction models (see PMID: 18511177 and PMID: 25403590). The only difference with our strategy is that we added one criterion to exclude models for recurrence and/or prebiopsy population. In addition, an alternative search strategy was conducted and identified almost 2000 papers, each of which have been systematically checked by two independent reviewers (MA and AL) and a sample by a third reviewer (KM) and only two papers have potential, however, full text was not available for further examination for one study (<https://doi.org/10.1016/j.juro.2018.02.1460>) and for the second one the article used data set to demonstrate how to build the risk model rather than developing a model to improve prostate cancer screening performance. The alternative search strategy is available upon request.
- a) With regards to Vickers paper that the reviewer referred to, it actually has similar Mesh terms that we used. For example, “Risk Assessment” and “Early Detection of Cancer” are included in Vickers paper as well as in our search strategy. We have not included the Vickers model as it requires serial measurements to assess PSA velocity and our search criteria has excluded it. We expand that in the first paragraph in the discussion section to be more clear.

- b) We can reassure that the review was conducted by a team that includes people with extensive experience of systematically searching and reviewing the literature.

Comment 4:

- While the authors state that the PRISMA statement was followed, there are some major omissions that should be part of a high quality systematic review.
 - a. The date of the search and exact combination of search terms for each search engine should be specified. In the current state, I would be unable to replicate the study.
 - b. How was Medline accessed? Through OVID, Web of Science or other?
 - c. Why were the articles limited to English? This is becoming an increasingly unjustifiable reason for exclusion, and many non-English speaking countries provide good studies given limited resources. Expanding this is worth considering.
 - d. Screening of articles was only performed by one reviewer – this should be performed by two reviewers, with discrepancies resolved after discussion or consultation of a third party.

Response:

- a) We stated the date of the search under “Data sources” and search strategy in “Methods section” as the following “All articles defined (published up to the end of January 2019)”. However, we changed it to this “All articles defined (published since the inception of the databases and up to the end of January 2019) to make it clear. The combination of search terms is provided as a supplementary file.
- b) Medline was accessed through OVID, and we state that in first line under Data sources and search strategy.
- c) We confirm that we have only included those studies reported in English. Also, we acknowledged it was one of the study limitations.
- d) The second reviewer (AL) has screened the results with any discrepancies resolved by the involving a third author (KM).

Comment 5:

- Statistically, it is stated “Due to the heterogeneity of the studies included, conducting a meta-analysis was not applicable.” However, there appears to be a prevalence meta-analysis performed for the AUC estimates. I am not sure how valid the multiple comparisons of the same databases/study populations with different variables is. I presume from the author affiliations that they are skilled in statistical principles, however if not, then perhaps consultation with a statistician with expertise in meta-analysis is warranted.
 - a. If prevalence or other meta-analysis was performed, what parameters and software was used?
 - b. Further to this, potential subgroup analysis may allow better explanation of heterogeneity (cancer detection rates 20-50% is a huge variation).

Response:

- We did not perform meta-analysis. We understand there is misunderstanding of Figure 2 that looks like we did perform a meta-analysis. Therefore, we have deleted the figure while the AUC's of the models are already presented in Table 3.

Comment 6:

- A key evolution from the screening trials and risk stratification is clinically significant cancer detection rate, can this also be provided for each study? This really is the key in guiding management. While I appreciate that the likely target here is the wider medical community and general practitioners/primary care physicians, perhaps it is worth the authors consulting an academic urologist to address the nuances in PSA-based prostate cancer detection and management.

Response:

- Whilst we agree that the ultimate aim of prostate cancer detection is to identify clinically significant prostate cancer, this is currently not optimised in community-based assessment. Our review reflects this and assess currently available approaches.

Comment 7:

- The introduction is appropriate in scope and length given the target general audience of BMJ Open.

Response:

- We thank the reviewer for pointing this out.

Comment 8:

- Were the PSA assays used the same? There are many different assays that can cause inter-test variability, so may be worth listing in a demographic or supplementary table.

Response:

- We have listed the PSA assays used in the included models in a supplementary table as the reviewer suggested.

Comment 9:

- Table 1 – median age should also be included.

Response:

- Median age was added to Table1.

Comment 10:

- Page 12 Lines 31-53 is difficult to follow.

Response:

- We tagged each model by its author for more clarity.

Comment 11:

- Page 16 Lines 7-9 “The event per variable (EPV) was lower than recommended (< 10) in the Babaian 44 study indicating inadequate power.” – is this a valid measure and if so, please provide supporting literature.

Response:

- We have provided supporting evidence (two citations).

Comment 12:

- As BMJ Open is an online journal, I would suggest colour coding in addition to the +/- for Table 6.

Response:

- We have colour coded Table 6 as the reviewer suggested.

Reviewer #2:

Comment 1:

- Overall, nice paper.

Response:

- We thank the reviewer for his feedback and his points addressed accordingly.

Comment 2:

- However, I think the authors need to be very careful about the message they are sending with this paper. The authors want to look at a 'non-clinical' risk calculator that does not include 'invasive' testing, and include the digital rectal examination (DRE) in this invasive category. They seem to justify this statement in the introduction because the DRE may cause anxiety, reference number 36. This reference actually does not state that the DRE may cause anxiety, in fact it states the opposite and references an earlier study demonstrating the DRE does NOT cause additional anxiety. This sentence in the introduction should be removed.

Response:

- We thank the reviewer for pointing this out. We have changed it to more appropriate references. However, as DRE is one of the common tests used for PCa diagnosis, we want to highlight the reason behind excluding it i.e. invasive/uncomfortable. Therefore, we have removed the old reference and added a paragraph in the introduction highlighting the limitations of both DRE and TRUS to support the claim.

Comment 3:

- This also raises the greater point of the use of DRE in prostate cancer risk stratification. It is essential that a DRE is performed in patients under investigation for prostate cancer, to suggest otherwise would not be good practice, which is implied by this study not including risk models using DRE. I think the only way to justify this study is to talk about the difficulty of an accurate DRE in the community setting, i.e. not from a urologist. I do not know the literature on this, but I think the authors need to think carefully about this before re-writing the introduction to justify not including DRE as a risk factor in a PCa risk model.

I feel the addition of a urologist to the paper, although not essential, would add a clinical opinion, which would be of great benefit to this work.

Response:

- We have added a new paragraph about the limitation of both DRE and TRUS in the introduction specifying that DRE is not applicable and less desirable.

Comment 4:

Methods:

- The authors do not state the statistical software and method used to create Figure 2, please add to methodology.

Response:

- We have removed Figure 2 for clarity, while the AUC's of the models are already showed in Table 3. The reason for removal can be found in reviewer 1's response.

Comment 5:

- The authors state that external validation is very important for a risk model and include this in the paper. "External validation was only carried out in one paper". However the authors have chosen not to include external validation studies in the analysis. How do the authors reconcile this? If there are external validation studies of the models included in this paper then the external validation studies should be mentioned.

Response:

- We do agree with the reviewer comment. We have searched both google scholar and Medline for any article(s) that cited each included model and we did not find any independent study for external validation. The search results are presented in supplementary table and the search method are added in the methodology section.

Comment 6:

Discussion:

- Very thorough discussion.

Response:

- We would like to thank the reviewer for finding the discussion part very thorough.

Comment 7:

- Please include in the limitations section that because you are searching for a non-clinical risk model for PCa, none of the models which include the PHI score are suitable as this would mean the addition of a biochemical test that is not routinely performed in clinical practice. This would have additional cost implications.

Response:

- We recognised that PHI is a useful marker but highlighted that two of the models included in our review do include PHI as they were identified by our search strategy. Also, According to NICE, the PHI can be run from same patient blood sample and can be conducted in a routine blood laboratory (see Diagnosis and monitoring of prostate cancer: PROGENSA PCA3 assay and the Prostate Health Index (PHI)). Moreover, a recent cost-analysis study showed that although PHI test is more expensive than PSA test, it could be reimbursed by the cost savings as a result of reducing unnecessary biopsies (PMID: 29980838).

Comment 8:

- I do feel as if the authors do not fully understand the management pathway of a patient under investigation for PCa. The use of a 'non-clinical' PCa risk calculator may fit into the clinical pathway, but the point it would fit in would have to be in the community to aid in the decision to refer a patient to a urologist. At which point improved risk stratification could then be undertaken following a DRE. The decision would then be made on whether to proceed to MRI, proceed directly to biopsy or to not perform biopsy. If an MRI is performed, the decision would then be made on prostate biopsy. Please include in the intro/discussion (as appropriate) where in the pathway a non-clinical model may be of use.

Response:

- We agree with the comment and have added two sentences under implications and future research paragraph 4.

Comment 9:

- The authors have recommended that a non-clinical risk model be made, and externally validated etc. Stating that "It is crucial to address these issues by identify all possible risk factors for PCa that are non-clinical, non-genetic, and easy to use and interpret." However, I am not sure there is a clinically unmet need for a "non-clinical" risk calculator that does not include DRE, so the authors will need to justify this with evidence from the literature (and perhaps the input of general practice and urology colleagues).

Response:

- We believe it is desirable to have a non-clinical risk model that can be used in primary care settings without DRE due to known limitations of DRE within primary care as described further in the additional text in the paragraph 5 in the introduction section.

Reviewer #3:

Comment 1:

- Well written paper and analyses.

Response:

- We thank the reviewer for his positive feedback.

Comment 2:

- Please add more clinical background to the introduction and discussion.

Response:

- Thank you for the suggestion. We have addressed this and added further detail within the introduction and discussion.

Comment 3:

- Please describe the limitations in DRE and TRUS.

Response:

- We added a new paragraph in the introduction to explain the limitations of both DRE and TRUS as suggested.

Comment 4:

- Please describe the additional role of multiparametric MRI in an initial phase of prostate cancer detection (after PSA, before biopsy), as this is non-invasive and non-genetic (and falls within the description of the title). Although mpMRI has obviously no role in the primary health care setting, but has a role in the hospital setting.

Response:

- We do agree that MRI increasingly has a role but currently not in primary care settings but as the reviewer suggested in the secondary settings and it is beyond the scope of the paper. However, the risk prediction in primary care will help screen those who actually need to have an MRI. We have added sentences to this effect in the discussion.

Comment 5:

- Please mention the ultimate goal of prostate cancer screening: to find intermediate and high risk prostate cancers (and not low risk prostate cancers that would not require treatment, but will give emotional burden to the patient once detected and generate unnecessary treatment in patients not coping with active surveillance treatment).

Response:

- Thank you. We have added the sentence as suggested in paragraph 4 in the introduction.

Comment 6:

- Please adapt your title to something like: 'prediction models for prostate cancer for primary health care: a systematic review', as this may better reflect the purpose and suggested use of the outcome of the review.

Response:

- Thanks for the suggestion. We agree and changed the title as suggested.

Reviewer #4:

Comment 1:

- This is a very nicely written article - I appreciated the opportunity to read it. The review was carefully undertaken and thoughtfully interpreted. I particularly enjoyed the Discussion and Conclusions, where the authors seem to be unconvinced by the current evidence on the available tests for community-based testing.

Response:

- We thank the reviewer for the encouraging comment.

Comment 2:

- My main concern is whether the article adds to the literature. In particular, there is so much heterogeneity between the studies, including PSA thresholds, study cohort definitions and biopsy protocols, that comparisons between the studies becomes increasingly uninformative. I realise that the change in AUCs is trying to provide more valid internal contrasts, however study heterogeneity may also be associated with those deltas.

Response:

- We do believe this article will provide some further information about prostate cancer risk prediction that can be used in primary care settings. With regards to heterogeneity, we do appreciate the comment, however, we did not perform any meta-analysis. We only looked at each study in a very detailed manner to see how well each model performs. We do understand that Figure 2 may confuse the reader and therefore, we deleted it and the AUCs are already presented in a Table 3.

Comment 3:

- I was less convinced about the importance of the population being biopsy-naive -- particularly when the current populations will increasingly have a large proportion of men having had a previous biopsy.

Response:

- Men who have been biopsied before are likely to have a raised PSA level or abnormal DRE as indicative of a biopsy (see PMID: 26332503). Including them could increase bias towards the PSA level. Moreover, the fact that some patients may get medicine to lower the PSA level which

ultimately would affect the outcome. Also, including men who had prebiopsy means including models that incorporated some variables related to the biopsy (such as number of biopsy cores) and biopsy itself is an invasive procedure. Such models are designed for those who are at persistent high risk despite previous negative biopsies (see PMID: 16643613). Those were the reasons behind excluding them.

Comment 4:

- Moreover, I was unclear why genetics should not be routinely included in community screening, particularly if such tests come down in cost. The authors may care to comment on these issues.

Response:

- We thank the reviewer for his comment. Genetic tests are usually conducted in specialist genetic service and it is not common practice in community settings, therefore, when a GP or a patient demand for a genetic test, a referral seemed appropriate. However, we do agree with the reviewer that if the cost comes down, such a test should be included in the community settings. We have added a sentence in the last paragraph in the discussion under implications and future research to reflect that.

Comment 5:

- There is an interesting issue with prostate cancer testing: should it be framed in terms of community testing or in terms of a clinical diagnostic pathway? For example, community-based testing should be inexpensive, however the choice of "screening" test affects *who* will be referred to a urologist, who may undertake an MRI, and may include further clinical information (e.g. second PSA test value, DRE, or prostate volume) to decide whether to undertake a biopsy. Decisions earlier in this testing and diagnostic pathway have resource and predictive implications for later in this pathway. The authors have set themselves firmly at the start of this pathway. Do these issues affect the interpretation of their findings?

Response:

- In the UK, if a man has a high PSA level, he will be invited to do the PSA test after a certain time and if the PSA level is still high, he will be referred to a clinician/urologist to do further examinations. The idea of community screening is to flag up men who are at higher risk and should be further investigated. Our literature search focused on the goal to find risk models that have better accuracy than PSA alone. As a result, we do not think that it should affect the diagnostic pathway.

Comment 6:

- Remarkably, I have little to criticise for the existing manuscript. Some of the English language could be improved. I have the following minor points.

Response:

- We thank the reviewer for his feedback and we addressed the points he made accordingly. The manuscript has been checked by English native speaker.

Comment 7:

- I was unclear about the interpretation of the phrase "evaluating the prospective risk" - please re-phrase for clarity.

Response:

- Thank you. We have changed it to be more clear. We deleted the word "prospective" and changed it to "evaluating prostate cancer risk". Changes were made both in the abstract and method sections.

Comment 8:

- The search terms seemed somewhat specific - what was the rationale for those terms, please?

Response:

- We used search terms similar to two previous systematic reviews on prostate cancer risk prediction models (see PMID: 18511177 and PMID: 25403590) and added one more criterion to exclude models for recurrence or include prebiopsy populations.

VERSION 2 – REVIEW

REVIEWER	Matthew Roberts University of Queensland, Australia
REVIEW RETURNED	10-Mar-2020

GENERAL COMMENTS	The authors are to be commended on the improvements made. The discussion does discuss the individual studies, however an overall assessment or recommendation based on the available literature (other than future directions) is lacking. I would suggest, for this to be a valuable reference for primary care physicians and health systems researchers/stakeholders, if the authors could make some suggestions in the conclusion or end of the discussion as to which currently available model they believe would be most beneficial for use in primary care. While I appreciate the discriminatory ability of these models is not perfect, they are an improvement on the currently used method (serum PSA). The balance of which model is most valid, free from bias and applicable to the UK and/or general population would be appreciated.
---

REVIEWER	Mark Clements Karolinska Institutet
	I am a co-investigator on the STHLM3 diagnostic trial. I have no financial conflict of interest associated with the STHLM3 test and no other competing interests.
REVIEW RETURNED	19-Mar-2020

GENERAL COMMENTS	1. The authors have provided a good revision of their article.
--

	2. The other reviewers and I were concerned about the sensitivity of the search criteria. However, the authors chose to not revise the search criteria. 3. Unfortunately, although the topic is timely and important, I remain unconvinced that the article adds sufficiently to the literature.
--	--

VERSION 2 – AUTHOR RESPONSE

Author Response to reviewer 1

Comment 1:

- The authors are to be commended on the improvements made.

Response:

- We thank the reviewer for his comment.

Comment 2:

- The discussion does discuss the individual studies, however an overall assessment or recommendation based on the available literature (other than future directions) is lacking. I would suggest, for this to be a valuable reference for primary care physicians and health systems researchers/stakeholders, if the authors could make some suggestions in the conclusion or end of the discussion as to which currently available model they believe would be most beneficial for use in primary care. While I appreciate the discriminatory ability of these models is not perfect, they are an improvement on the currently used method (serum PSA). The balance of which model is most valid, free from bias and applicable to the UK and/or general population would be appreciated.

Response:

- We agree with the reviewer. We have added a sentence at the end of discussion highlighting the best current available model that has the potential to be used in the primary care setting.

Author Response to reviewer 4

Comment 1:

- The authors have provided a good revision of their article.

Response:

- We thank the reviewer for his comment.

Comment 2:

- The other reviewers and I were concerned about the sensitivity of the search criteria. However, the authors chose to not revise the search criteria.

Response:

- We did not ignore the advise about search criteria. We therefore cross checked our search terms strategy to ensure that we did not miss any eligible studies by conducting another broad search strategy that yielded around 2000 potential papers but when reviewed against our explicit criteria we did not find any further eligible studies.

Comment 3:

- Unfortunately, although the topic is timely and important, I remain unconvinced that the article adds sufficiently to the literature.

Response:

- We respect but disagree with the reviewer's comment. We undertook this review because we noticed a number of review articles that have not actually highlighted or focused on the use of risk prediction models in primary care. Therefore, we remain convinced that our article does add significantly to the published literature.